# Enantioselective Heck/Tsuji−Trost reaction of flexible vinylic halides with 1,3-dienes

Li-Zhi Zhang [1,5], Pei-Chao Zhang [2,5], Qian Wang[3], Min Zhou [1] ✉ & Junliang Zhang [3,4] ✉

The enantioselective domino Heck/cross-coupling has emerged as a powerful tool in modern chemical synthesis for decades. Despite significant progress in relative rigid skeleton substrates, the implementation of asymmetric Heck/cross-coupling cascades of highly flexible haloalkene substrates remains a challenging and and long-standing goal. Here we report an efficient asymmetric domino Heck/Tsuji−Trost reaction of highly flexible vinylic halides with 1,3-dienes enabled by palladium catalysis. Specifically, the Heck insertion as stereodetermining step to form $\eta^3$ allyl palladium complex and in situ trapping with nucleophiles enable efficient Heck/etherification in a formal (4 + 2) cycloaddition manner. Engineering the Sadphos bearing androgynous non-$C_2$-symmetric chiral sulfinamide phosphine ligands are vital component in achieving excellent catalytic reactivity and enantioselectivity. This strategy offers a general, modular and divergent platform for rapidly upgrading feedstock flexible vinylic halides and dienes to various value-added molecules and is expected to inspire the development of other challenging enantioselective domino Heck/cross-couplings.

Catalytic asymmetric domino Heck/cross-coupling in the past forty years were broad attraction and applications in the functionalization of C−C π-Bonds[1–5]. Relying on a relatively rigid skeleton substrate that is provided in aryl halides, these versatile domino reactions proved their powerfulness allowing high control in regio-, diastereo- and enantio-selectivities (Fig. 1a). In contrast to the underdeveloped highly flexible haloalkene substrates[6–9], the substrates of rigid skeleton frequently exhibited enhanced conformational stability involving elementary reactions, to make the reaction more beneficial to generate the desired product and inhibit the side reaction[10]. Indeed, as one of the elegant reactions, the domino Heck/Tsuji-Trost reactions[11–13] would permit the formation of multiple stereocenters in mono- and polycycles with high atom- and step-economic efficiency (Fig. 1b)[14–19]. In this regard, the enantioselective formal Heck/amination[20–25], Heck/etherification[26–28], and Heck/alkylation[29–31] with 1,3-dienes were established by Shibasaki, Luan, Gong, Overman, Han, and Zhang, independently, opening a new

era for asymmetric domino Heck/functionalization of conjugated dienes with rigid ambiphilic substrates. Specifically, pioneering studies were disclosed by Shibasaki[28] in 1991. Subsequently, Overman[24] group reported the enantioselective total synthesis of the fungal natural product (−)-spirotryprostatin B in 2000. With the development of novel chiral ligands, by utilizing the BINOL-based phosphine ligand, Gong[18] described elegant enantioselective redox-neutral difunctionalization of dienes in 2015. More recently, we[20–22] also developed the use of adaptive Sadphos ligand, enabling this cascade pathway through a stereoselective olefin insertion.

According to these seminal reports, which suggest: (1) a satisfying enantioselective protocol for the highly flexible haloalkene substrates and homologs (Fig. 1a) is especially challenging and still waiting to be developed[8]; (2) the orderly activation of reactive site requires precise control at every stage in catalytic asymmetric cascade process while avoiding transition metal-catalyzed direct allylic functionalization via

[1]School of Ethnic Medicine, Yunnan Minzu University, Kunming, Yunnan, China. [2]The Center for Basic Research and Innovation of Medicine and Pharmacy (MOE), School of Pharmacy, Second Military Medical University (Naval Medical University), Shanghai, P. R. China. [3]College of Chemistry and Life Science, Advanced Institute of Materials Science, Changchun University of Technology, Changchun, P. R. China. [4]Department of Chemistry, Fudan University, Shanghai, P. R. China. [5]These authors contributed equally: Li-Zhi Zhang, Pei-Chao Zhang. ✉e-mail: zhouminynun@163.com; junliangzhang@fudan.edu.cn

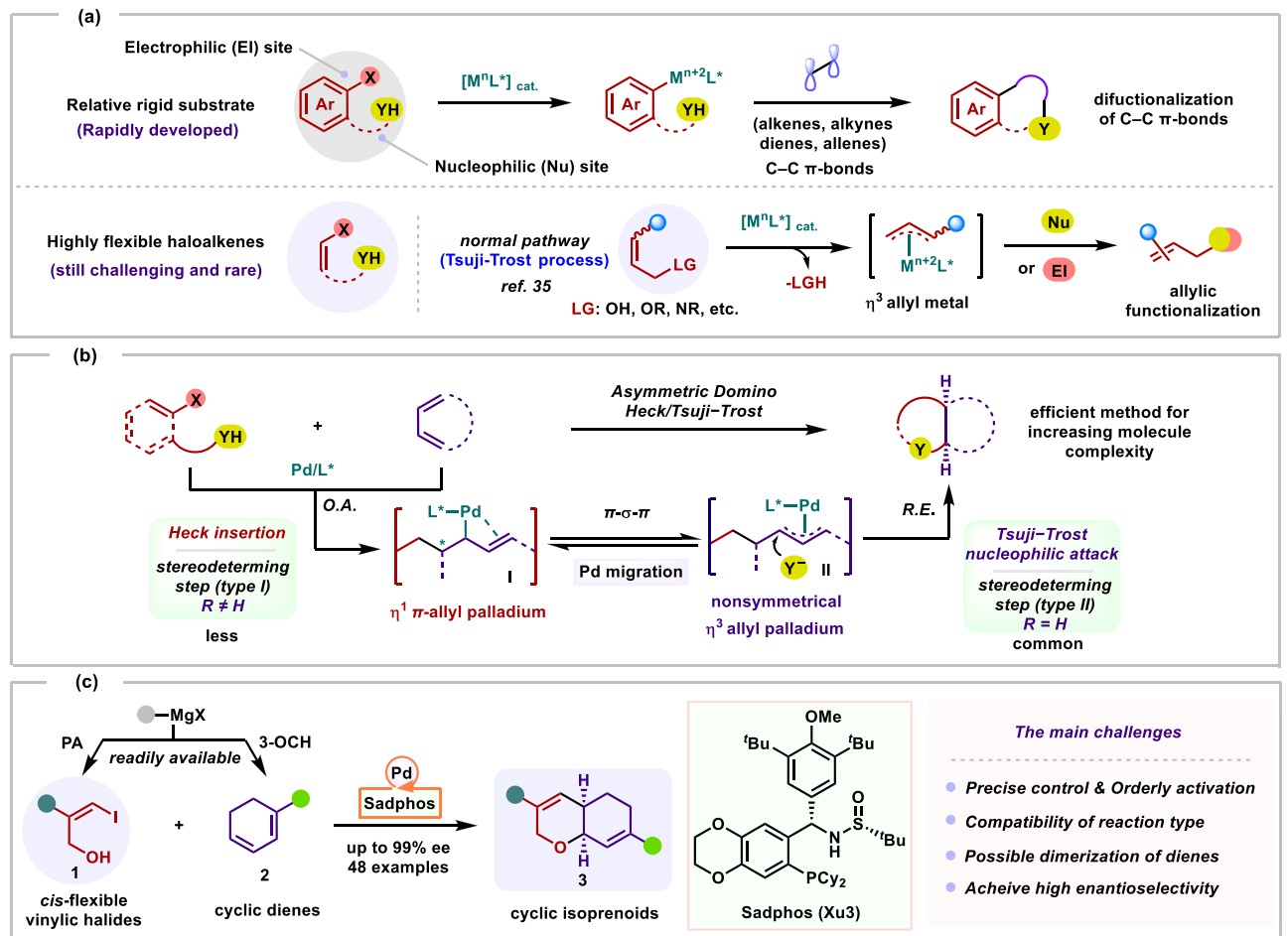

**Fig. 1 | Catalytic asymmetric domino Heck/Tsuji−Trost reactions. a** Asymmetric domino Heck/cross-coupling of rigid & flexible substrates. **b** Asymmetric domino Heck/Tsuji−Trost reactions with 1,3-dienes. **c** Asymmetric protocol for the highly flexible haloalkenes.

Tsuji-Trost reaction (Fig. 1a)[32–35]; (3) the traditional approach to such stereodetermining step relies on Tsuji−Trost nucleophilic attack step rather than Heck insertion step (Fig. 1b).

As part of our ongoing research into transition-metal/Sadphos-catalyzed[36,37] asymmetric annulation/cyclization reaction[38–40], herein, we envisaged that ambiphilic halogenated allylic alcohols **1** with readily available cyclic 1,3-dienes[12,13,41–43] **2** via the more challenging asymmetric domino Heck/Tsuji-Trost reaction to produce the enantioenriched sp³-rich cyclic isoprenoids (Fig. 1c). If successful, a variety of valuable chiral functional cyclic isoprenoids (Fig. 2a) could be easily prepared, which are key structural motifs[44] of numerous natural product family, pharmaceutical agents, and carbohydrates but remain challenging to access via asymmetric catalysis. Besides, several challenges would be encountered in this scenario: (1) Unactivated allylic alcohol substrates **1** may be directly activated via Tsuji-Trost reaction leading to electrophilic π-allyl palladiumintermediates[45–47]. (2) How to get high regioselectivity and enantioselectivity via the key stereodetermining step of Heck insertion[48]. (3) As yet, the development of catalytic asymmetric reaction with readily available and ambiphilic vinylic halides **1** has not been explored. Actually, we propose that the chiral ligand is crucial for overcoming these challenges.

## Results and discussion

With these considerations in mind, an initial attempt that the Xu-Phos (**Xu1**, one family member of Sadphos) could indeed enable the catalytic asymmetric domino Heck/Tsuji-Trost model reaction of flexible halogenated allylic alcohol **1a** with conjugated dienes **2a** or **2a´** to

access chiral sp³-rich cyclic isoprenoids (Fig. 2b). It's worth noting that the cyclic diene **2a** give the desired product in 23% yield with 81% ee, while the acyclic 1,3-diene **2a´** led to higher yield but with almost no ee, indicating that the stereodetermining step for this reaction is attributed to the Heck insertion step. And, these cascade reactions occurred chemo-, regio- and enantio- selectively at the less-hindered olefin of diene. To our delight, amide-type solvents and silver salt as the base could lead to desired product **3aa** up to 96% ee (Supplementary Fig. 2 and Supplementary Fig. 3). In addition, switching the counterion of palladium catalyst precursor from acetate to pivalate is beneficial to this transformation (Supplementary Fig. 5). With these preliminary results, we then turned our investigation on the asymmetric domino Heck/Tsuji-Trost reaction of **1a** with cyclic 1,3-dienes **2a** by using Pd(CO₂ᵗBu)₂ as a precatalyst and Ag₂SO₄ as the base in N,N-dimethylacetamide (DMAc) at 70 °C. A series of commercially available chiral rigid-flexible ligands (DIOP, Trost's ligand, BOX, Josiphos, Segphos, BINAP and other family members of Sadphos), which also have shown good performance in asymmetric π-allylpalladium chemistry, were first investigated (Fig. 2c and Supplementary Fig. 1), these results once again revealed the fact that adaptive Sadphos ligand is the key involved in regulating the domino Heck/cross-coupling. Previous studies have suggested that modifications to the electron-nature of the ligand backbone can influence both the catalytic activity and the enantioselectivity[49,50], Xu-Phos (**Xu2 − Xu5**) bearing electron-donating group on the benzene backbone were then synthesized and subjected to the reaction. To our delight, employing **Xu3** as ligand, the yield was indeed significantly improved

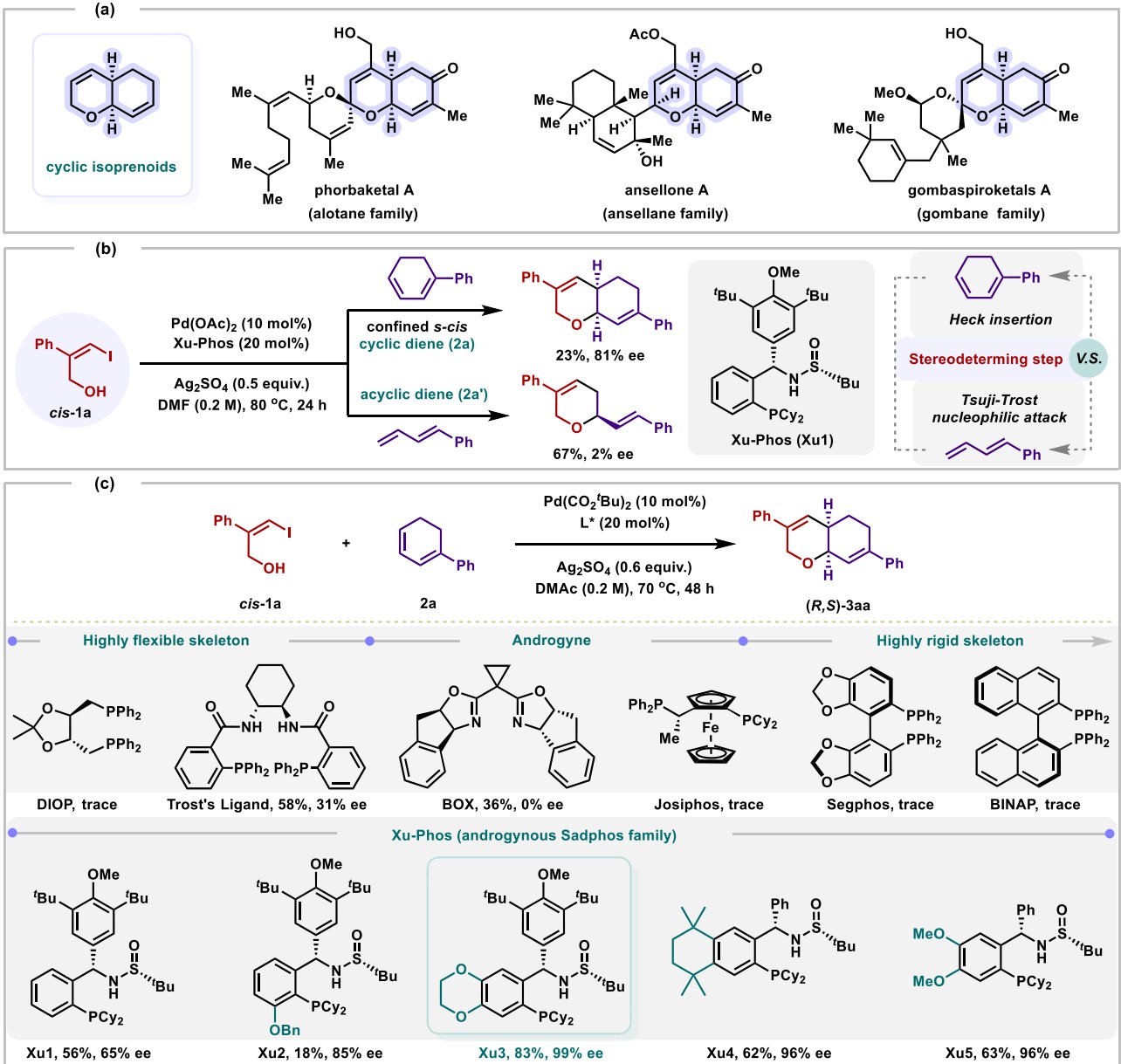

**Fig. 2 | Ligand enabled catalytic asymmetric domino Heck−Tsuji−Trost reaction of flexible vinylic halides with 1,3-Dienes.[a-d]. a** Natural product featuring a cyclic isoprenoid skeleton. **b** Preliminary attempt. **c** Screening of chiral ligands. [a] **1a** (0.1 mmol), **2a** (0.4 mmol), palladium catalyst (10 mol%), ligand (20 mol%), silver salt (0.6 equiv), solvent (0.2 M), Ar, 70 °C, 48 h. [b] Yields are determined by GC analysis using anisole as an internal standard. [c] Isolated yield after flash-column chromatography. [d] Determined by HPLC analysis.

from 56% to 83% with the enantioselectivity increased from 65% to 99% ee.

With the optimal reaction conditions in hand, the generality of substrates in this asymmetric domino Heck/Tsuji-Trost reaction of ambiphilic and flexible vinylic halides **1** with conjugated dienes **2** was then investigated as depicted in Figs. 3 and 4. Notably, flexible vinylic halides **1** are easily synthesized by the nucleophilic addition of propargyl alcohol (PA), with a large range of substituted alkenes[51]. The structure and configuration of (*R,S*)-**3aa** was unambiguously determined via its X-ray analysis (CCDC: 2323645). Initially, the results demonstrated that vinylic halides **1** bearing halogens (fluorine, chlorine), electron-donating groups (tertiary butyl, methyl, methoxyl) at various positions of the phenyl ring were compatible, delivering corresponding products **3aa**–**3ag** in good to high yields with 84–99% ee. To our delight, various substituents and functional groups on the flexible vinylic halides **1** could be tolerated. For example, 2-naphthyl, 2-

allyl, terminal *n*-butenyl, and *n*-pentenyl could also produce the corresponding target products **3ah**–**3ak** in high yields with 93–99% ee. It is particularly worth mentioning that a series of more flexible straight-chain alkyl, branched-chain alkyl, and cycloalkyl all can deliver the cyclic isoprenoids **3al**–**3ax** in good to excellent yields with 85–97% ee as a single regioisomer and diastereoisomer. The bicyclic isoprenoid compound **3** shares the core structure with several monoterpene lactones, making it a promising synthetic intermediate for the production of these bioactive natural substances[44].

On the other hand, conjugated dienes **2** can be readily synthesized via the 1,4-dehydration of allylic alcohols. The *s-cis* conformation lockdown of the C=C bonds greatly aids in the formation of π-allylpalladium(II) complexes, leading to a decrease in the activation entropy. (Fig. 4)[52]. The applicability of this protocol toward various substituents and functional groups on the cyclohexadienes scope was investigated. For instance, substituents such as fluorine, chlorine, methyl, methoxyl,

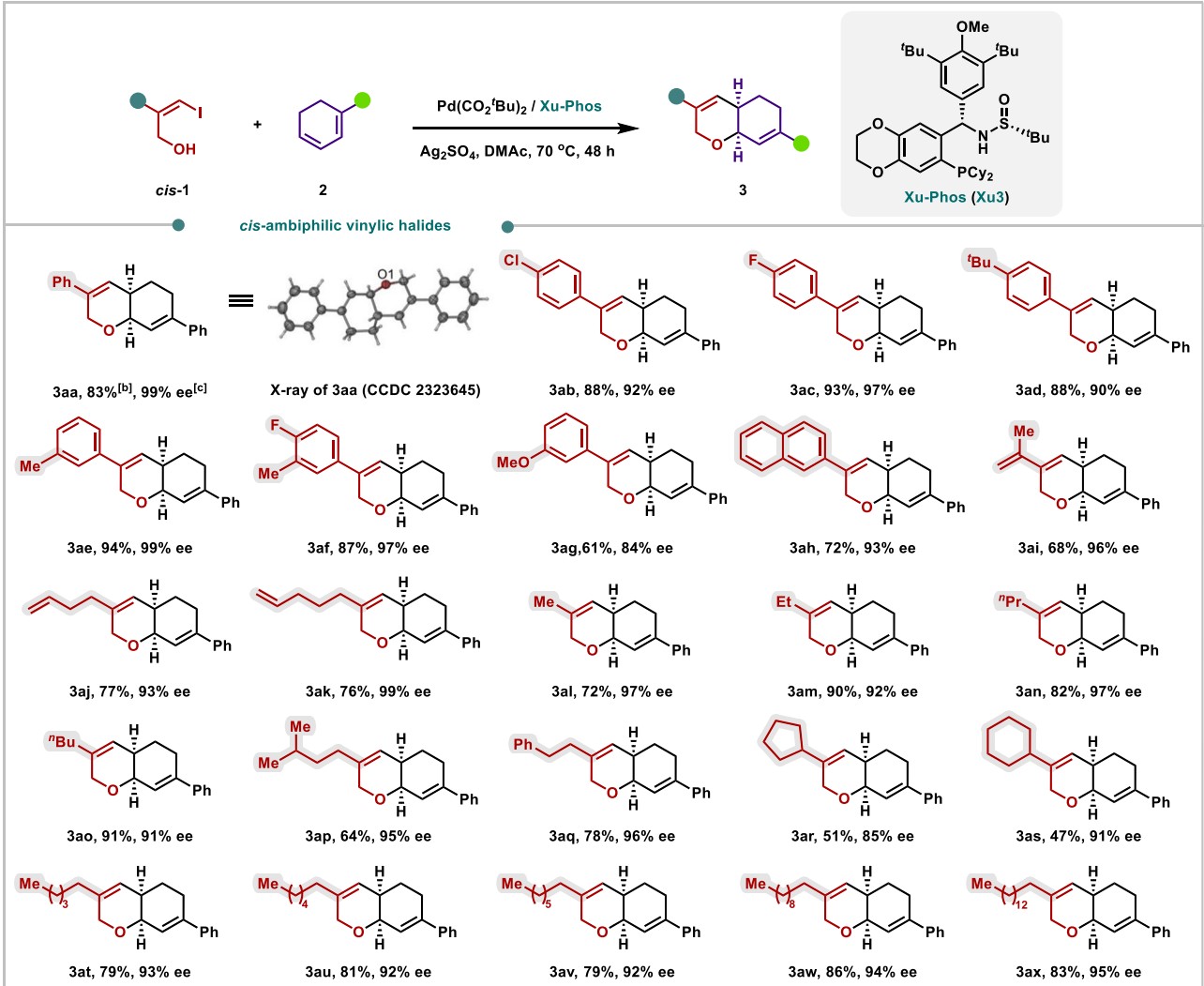

**Fig. 3 | Scope of the asymmetric domino Heck/Tsuji-Trost reaction of 1 with cyclohexadienes 2.** The yields reported represent single runs and have not been reproduced in this work.[a] [a] **1** (0.2 mmol), **2** (0.8 mmol), Pd(CO$_2$$^t$Bu)$_2$ (10 mol %), (Sc,Rs)-**Xu3** (20 mol%), Ag$_2$SO$_4$ (0.6 equiv), DMAc (1 mL), Ar, 70 °C, 48 h.

trifluoromethoxy, trifluoromethyl, and trimethylsilyl on the aryl moiety of 1-aryl-cyclohexa-1,3-dienes **2** are compatible, delivering the desired **3ba–3bj** in 57–93% yields with 90–99% ee. Moreover, 1-naphthyl, 2-naphthyl, dioxa-phthyl, 5-benzothienyl, 3-thienyl, 1-vinyl, 1-phenylethy-nyl, and n-butyl derived cyclohexa-1,3-dienes could also produce **3bk–3bs** in 52 − 94% yields with 83–97% ee. Specifically, the 1-vinyl and 1-phenylethynyl groups act as versatile handles for subsequent modifications of the bicyclic rings. And 1-vinylcyclohexadiene, which functions as a conjugated triene containing mono-, di-, and trisubstituted olefins, selectively cyclized at the cyclic and less-substituted olefin portion. This selectivity is likely due to the more effective orbital overlap of the cyclic diene. With the derivatives of pharmaceuticals (menthol and perillyl alcohol) as the dienes, the corresponding products **3bt** and **3bu** could be obtained in moderate yields with excellent diastereoselectivity.

To demonstrate the practical utility of our protocol, a gram scale reaction was carried out under standard reaction conditions, furnishing 1.14 g of **3aa** in 79 % yield with 99% ee (Fig. 5a). Moreover, the unsaturated bonds present in the cyclic products **3** offer opportunities for further diverse modifications. For instance, the selective dihydroxylation of **3aa** with K$_2$OsO$_4$ delivered the target product **4** in a 69% yield with 99% ee. The hydrogenation of **3aa** in the presence of Pd/C furnished octahydro-2H-chromene product **5** in 87% yield with 99% ee.

The selective difluorocyclopropanation of **3aa** led to the highly functionalized product **6** in 74% yield with 96% ee. The selective epoxidation of the two olefin moieties of **3aa** with m-CPBA delivered the target products **7** in 77% yield with 99% ee. In light of the structures of the chiral Pd/Sadphos catalyst[37] and the product **3**, a possible catalytic chirality-induction model was proposed for the reaction (Fig. 5b). The 8-membered ring of O,P-chaleting complex, the less-hindered olefin coordinate to the Pd(II) center and the Re-face of alkene is shielded by the 3,5-ditert-butyl-4-methoxy-phenyl group of the ligand leads to intermediate **Int-l**. Because of these, the syn-migration insertion of 1,3-diene **2** into the C−Pd bond would deliver a palladium complex **Int-ll**. The intramolecular nucleophilic attack takes place at the Si-face to form the cis-product.

In summary, we have developed a highly chemo-, regio-, and enantio-selective palladium-catalyzed asymmetric domino Heck/Tsuji-Trost reaction of flexible halogenated allylic halides with cyclic 1,3-dienes. This reaction serves as a promising tool for the modular synthesis of enantioenriched sp³-rich cyclic isoprenoids. The androgyne **Xu-Phos** ligand plays a crucial role in regulating catalytic activity and selectivity of this domino Heck/cross-coupling. Further studies will focus on the application of Sadphos in asymmetric metal catalysis, particularly in domino Heck/Tsuji-Trost reactions involving other challenging reactions and substrates.

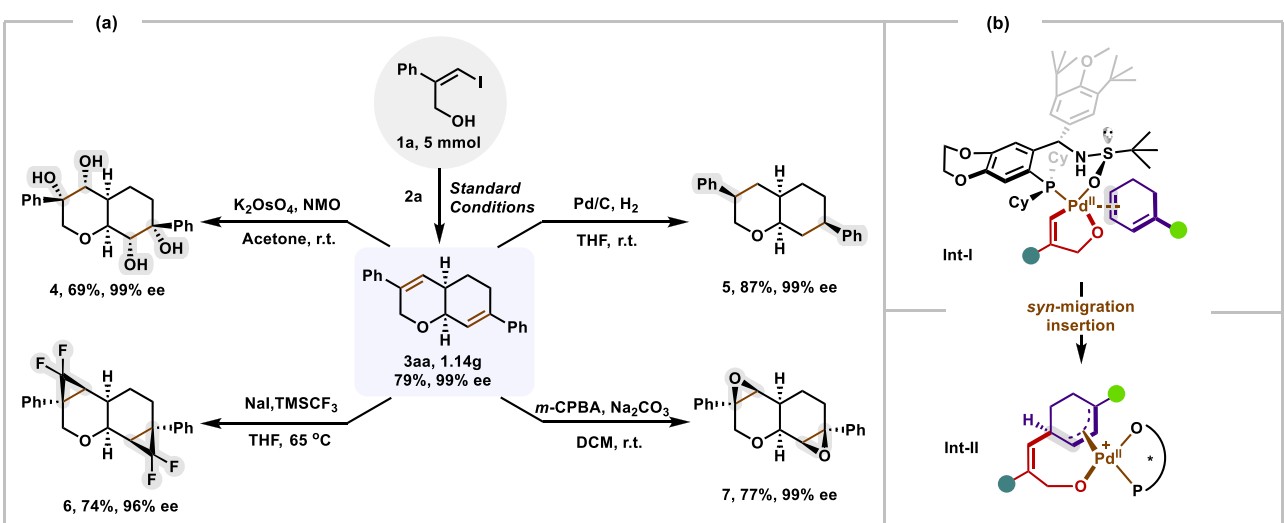

**Fig. 4 | Variation of conjugated cyclohexadiene 2.** The yields reported represent single runs and have not been reproduced in this work.[a] [a] **1** (0.2 mmol), **2** (0.8 mmol), Pd(CO₂ᵗBu)₂ (10 mol %), (*Sc,Rs*)-**Xu3** (20 mol%), Ag₂SO₄ (0.6 equiv), DMAc (1 mL), Ar, 70 °C, 48 h.

**Fig. 5 | Synthetic transformations and plausible mechanism. a** Gram-scale reaction and Functional transformations. **b** Plausible asymmetric induction model.

## Methods
### General procedure for synthesis of 3aa-3bu
To a sealed tube was added Palladium pivalate (10 mol%, CAS: 106224-36-6, *Bide*) and **Xu3** (20 mol%) in 1 mL dry DMAc and stirred at room temperature for 1 h under argon atmosphere. Then,

**1** (0.2 mmol, 1.0 eq), **2** (0.8 mmol, 4.0 eq), and Ag₂SO₄ (0.12 mmol, 0.6 equiv) were added to the tube under argon atmosphere and stirred at 70 °C for 48 h. After the reaction was complete (monitored by TLC), diluted with saturated salt water and EA, then extracted with EA (twice), and dried over anhydrous Na₂SO₄, the solvent was

removed under reduced pressure. The crude product was purified by column chromatography (*n*-Hexane/EA, 50:1 to 30:1) to give **3** as a white solid or colorless liquid.

## Data availability

All data supporting the findings of this study are available within the article and its Supplementary Information. Crystallographic data for the structures reported in this article have been deposited at the Cambridge Crystallographic Data Center (CCDC), under deposition number 2323645 (**3aa**). Copies of the data can be obtained free of charge via https://www.ccdc.cam.ac.uk/structures. Data relating to the characterization data of materials and products, general methods, optimization studies, experimental procedures, mechanistic studies, and NMR spectra are available in the Supplementary information. All data are also available from the corresponding author upon request.

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

## Acknowledgements
We gratefully acknowledge the funding support of National Key R&D Program of China (Grant 2021YFF0701600) to J.Z., the National Nature Science Foundation of China (Grants 32060101, 22031004, 21921003 and 22401293) to M.Z., J.Z., and P.-C.Z., the Shanghai Municipal Education Commission (Grant 20212308) to J.Z., and the School Youth Initiation Foundation (Grant 2023QN024) to P.-C.Z. We greatly appreciate Yanfei Niu and Prof. Xiaoli Zhao at East China Normal University for their kind help with X-ray single crystal structural analyses.

## Author contributions
J.Z., P.-C.Z., and M.Z. conceived the project, analyzed the data, and wrote the paper. L.-Z.Z. and P.-C.Z. performed the most of experiments. Q.W. helped in the synthesis of substrates. All authors discussed the results and commented on the paper.

## Competing interests
The authors declare no competing interests.
