## [Transparent Peer Review file · Nature Communications]

Enantioselective Heck/Tsuji–Trost Reaction of Flexible Vinylic Halides with 1,3-Dienes

Corresponding Author: Professor Junliang Zhang

Version 0:

Reviewer comments:

Reviewer #1

(Remarks to the Author)

This manuscript by Zhang and co-workers reports an asymmetric tandem Heck/Tsuji-Trost reaction catalyzed by palladium, utilizing highly flexible ethylene halide and 1,3-dienes, achieving excellent regio-, diastereo-, and enantioselectivity. The SadPhos ligand developed by the research group plays a crucial role in attaining outstanding catalytic activity and enantioselectivity, albeit a similar strategy has been previously demonstrated in their earlier research (refs 18-20). Overall, the current research is of high importance, and is recommended for publication after the following issues being addressed.

(1) The authors suggest avoiding direct functionalization of alkenyl catalyzed by transition metals. Indeed, the product of such direct functionalization would result in a rigid four-membered ring, which is energetically unfavorable. Have the authors observed any formation of this four-membered ring product?

(2) The Pd and ligand loadings are notably high. Did the authors attempt to reduce the amount of catalyst used, and if so, were there any effects on the yield and enantioselectivity?

(3) When using Pd(CO₂tBu)₂ instead of Pd(OAc)₂, the yield increased from 32% to 62%. How do the authors account for this improvement? In cases of low yield, is it due to unreacted substrate?

(4) In Fig. S1, the yield of Xu8 is comparable to that of Xu1, yet the enantioselectivity is significantly higher. What prompted the modification of the catalyst structure based on Xu1 rather than Xu8?

(5) Is it necessary for flexible ethylene halide 1 to bear substituent groups in Fig. 3? Have the authors experimented with substrates lacking substituents, and if so, what were the results? Can the authors produce seven-membered ring products? The authors posit that stereoselectivity arises from the Heck reaction step. Have they tried using acyclodienes? Can enantioselectivity and diastereoselectivity be controlled in such cases?

(6) In the HPLC analysis, the racemate of compound 5 does not appear as a single compound. Additionally, in the NMR spectra, the solvent peaks for compounds 3ba and 4 are excessively high.

(7) All 2D spectra lack peaks and resolution.

Reviewer #2

(Remarks to the Author)

The authors describe an asymmetric heteroannulation reaction of vinylic halides with 1,3-cyclohexadienes. This reaction makes use of chiral phosphine/sulfinamide ligands, which are effective at inducing asymmetry in the reaction with cyclohexadienes. The authors show that acyclic dienes are ineffective in the reaction. Among the vinylic halides, a range of substitution works well in the 2-position, including both electron-rich and electron-poor aryl substitution and alkyl groups. A similar range of substitution is well-tolerated among the cyclohexadienes, although in neither case is there variation in the position of substituents, and the demonstration of functional group tolerance is largely lacking (the authors show that thiophene, benzothiophene work, but no nitrogen functionality is present throughout the scope). The reaction scales with comparable reactivity (79% vs. 83% yield), and the authors show some typical reactions for elaboration of the alkenes in the products. An asymmetric induction model is provided, although without experimental support.

Overall, this is a nice demonstration of asymmetric catalysis for the synthesis of interesting products. For the scope, the e.e. values are generally quite good, with a decent scope, and they map directly onto bioactive scaffolds of interest. While there have been several reports of enantioselective heteroannulation, this is to my knowledge the first with non-aromatic bifunctional reagents, which is a nice synthetic advance. However, despite the number of substrates, overall the substrate

scope is narrow. There is limited demonstration of functional group tolerance, and it is unclear whether the method can be extended to other cyclic dienes. The SI is generally good and quite detailed, but there are some issues with the reported HPLC traces that need to be addressed (see below for details). Also, the authors should indicate both in the substrate scope tables and SI that the yields reported represent single runs and have not been reproduced.

Specific suggestions:

- 1) Page 1, line 40-41: "seminal researchs" is misspelled, and would be more appropriate to say "seminal reports;" also the sentence appears to be a fragment and not a complete sentence as written. If the authors intend to use periods between the numbered items in this paragraph, they should capitalize the first word and make sure that they stand alone as sentences. I personally think it would be sufficient to use a semicolon between the numbered items.
- 2) Fig. 1A-B: The wording " η^3 π -allyl" is redundant; the authors should either say " η^3 allyl" or " π -allyl"
- 3) Fig. 4: From Fig. 2b, it seems that the reaction is limited to cyclic dienes for the purposes of asymmetric induction. However, the authors only show 1,3-cyclohexadienes in their scope, and there is no evidence of other cyclic dienes in the SI either. Will this reaction work with cyclopentadienes or cycloheptadienes? The authors should include these results to provide a better sense of the limitations (or potential diversity) of the substrate scope.
- 4) Fig. 5B: The authors propose a model for asymmetric induction, but without any direct experimental or computational support. They rely on prior work with different systems (most of the provided models in the ACR paper use olefins as the bound substrates, and many of those are thin on experimental support as well). It would be helpful for the authors to either provide more support for their proposed model (either through additional experimentation/computational study) or to use "plausible asymmetric induction model" instead of "proposed." In reality, this contributes little to what is otherwise an entirely synthetic paper, and could be left out.
- 5) References: While the authors' focus is on asymmetric catalysis, it would be highly relevant for them to include references to work from the Richard Larock group and Keary Engle group on olefin heteroannulation reactions with non-aromatic bifunctional coupling partners, which provide reactivity precedent for the authors' present asymmetric catalysis work with these types of coupling partners.
- 6) SI: Some of the racemic HPLC traces are not clean, with multiple impurities present near the peaks of interest; this could represent impurities that lie under the peaks of interest in the HPLC traces of enantioenriched product, leading to inaccuracies in the reported e.e. values. The authors should address these discrepancies. The relevant compounds are: 3ah, 3ak, 3bk (in this case, the peak integration of the racemic sample is 48:52, outside the acceptable error for a pure, racemic sample), 3bp, 3bs (peak integration of racemic sample is 43:57, with clear impurity peaks under the desired peaks),
- 7) SI: For compound 3an, there is significant overlap between the enantiomer peaks in the HPLC traces, making it very difficult to be confident in the integrations (and thus, reported e.e. values) of the enantioenriched traces. The authors should identify a different separation method in order to properly resolve the peaks of interest, and update their reported e.e. values accordingly.

Reviewer #3

(Remarks to the Author)

In this manuscript, Junliang Zhang and coworkers describe an enantioselective Heck/Tsuji-Trost domino reaction. The novelty of this protocol is firstly the incorporation of acyclic, flexible vinyl halides as substrates. Secondly, it is the Heck carbometalation that forms the stereo-determining step (the source of enantioselectivity) rather than the Tsuji-Trost allylation (that is widely explored). These are two quite remarkable points. High degrees of enantioselectivity are obtained and the versatility is documented by a variety of substrates.

Despite the progress the authors made in "acyclic stereoselection", a final challenge remains to be tackled (in the future): the step from the cyclic 1,3-diene to an acyclic 1,5-disubstituted 1,4-diene!

Nevertheless, the manuscript at hand discloses an innovative, valuable procedure. Therefore, I strongly recommend acceptance of the manuscript.

Some alterations are necessary:

- The term "tandem" is not adequate in this context and should be replaced by "domino", as explained by Lutz Tietze. In a tandem, a simultaneous motion occurs. This is not the case in the reactions at hand, where one step is prerequisite to the next. I also recommend citation of the comprehensive overviews: L.-F. Tietze, G. Brasche, K. M. Gericke, Domino Reactions in Organic Synthesis, Wiley-VCH: Weinheim, Germany 2006; b) L.-F. Tietze (Ed.), Domino Reactions: Concepts for Efficient Organic Synthesis, Wiley-VCH, Weinheim, Germany 2014.

- To me, parts of Fig. 1 are not clear. Fig. 1a, the second line simply illustrates the Tsuji-Trost reaction. Why is ref. 17 given here – instead of citations of comprehensive reviews by Barry Trost?

- Also Fig. 1b: Where are fragments and residue R (second line, in brackets) in the final product??

- Abstract: line 18 should read "as" instead of "of" (stereodetermining step).

- page 5: Fig.4 and line 114: menthol and perylic alcohol (no capitals)

In summary: I recommend acceptance of the manuscript after alterations, additions and corrections as outlined above.

Version 1:

Reviewer comments:

Reviewer #1

(Remarks to the Author)

The authors have made satisfactory revisions to their work, and I am pleased to reiterate my previous decision to fully

support its publication in Nature Communications. I am confident that it will make a positive contribution to the field.

Reviewer #2

(Remarks to the Author)

Thank you to the authors for thoughtfully addressing the concerns I had raised previously. I have no further suggestions and support its publication.

Reply to the comments and suggestions of the reviewer #1:

This manuscript by Zhang and co-workers reports an asymmetric tandem Heck/Tsuji-Trost reaction catalyzed by palladium, utilizing highly flexible ethylene halide and 1,3-dienes, achieving excellent regio-, diastereo-, and enantioselectivity. The SadPhos ligand developed by the research group plays a crucial role in attaining outstanding catalytic activity and enantioselectivity, albeit a similar strategy has been previously demonstrated in their earlier research (refs 18-20). Overall, the current research is of high importance, and is recommended for publication after the following issues being addressed.

Questions and Comments:

- (1) The authors suggest avoiding direct functionalization of alkenyl catalyzed by transition metals. Indeed, the product of such direct functionalization would result in a rigid four-membered ring, which is energetically unfavorable. Have the authors observed any formation of this four-membered ring product?

*Reply: We thank you for the high comments and good suggestions, which encourages us to conduct more comprehensive and systematic domino Heck/Tsuji-Trost reaction. That's a good question and we're sure that some readers also have the same question. Under standard reaction conditions, flexible vinylic halides **1a** without conjugated dienes **2a** was then investigated, we haven't found the intramolecular rigid four-membered ring product, which should be unstable (acid sensitive vinyl ether moiety in rigid four-membered ring) even it is formed. According to our results and previous studies (ref. 32-35), at present we suspect that unactivated allylic alcohol substrates **1** may be direct activated via Tsuji-Trost reaction leading to electrophilic π -allyl palladium intermediates, during the leaving group (OH) is expelled. We have rewritten this assumption to make it clearer in Manuscript. (See, the second paragraph: According to these seminal reports, which suggest:;(2).....while avoiding transition metal-catalysed direct allylic functionalization via the Tsuji-Trost reaction).*

**Tsuji-Trost process:**
- (2) The Pd and ligand loadings are notably high. Did the authors attempt to reduce the amount of catalyst used, and if so, were there any effects on the yield and enantioselectivity?

Reply: According to your nice suggestions, we screened the amount of the Pd and ligand loadings. Similar to our previous work (Chem. 2022, 8, 836), when the amount of catalyst is reduced, the yield exhibits significant variation. The control experiments show that the ratio of Pd:ligand >1:1 is ok

for the yield and enantioselectivity, but taking accounts of Sadphos is easy to make in large scale, we chose the Pd/L* catalyst loadings of 10/20 mol% respectively. The results were discussed in the revised Supplementary Information. (SI, Figure S7).

Entry	Pd:L (%)	GC Yield (%)	Ee (%)
1	10:10	71	94
2	10:12	80	97
3	10:20	83	99
4	5:10	54	97
5	6:12	60	96

(3) When using Pd(CO₂tBu)₂ instead of Pd(OAc)₂, the yield increased from 32% to 62%. How do the authors account for this improvement? In cases of low yield, is it due to unreacted substrate?

Reply: Thank you for pointing this out. This is a very interesting result. On the one hand, we supposed that Pd(O₂C^tBu)₂ has better solubility than Pd(OAc)₂ to facilitate in situ generation of Pd(0). On the other hand, the use of a bulky tert-butyl ester as a ligand is advantageous for the reduction elimination processes, and accelerate the initiation Heck step of this reaction. (Chem. Rev. 2003, 103, 1979).

In cases where the yield is low, substrate 1 has consistently been found to decompose. We have provided this comment to rationalize this observation in Supplementary Information. Hope it's clear. (SI, Page S5).

(4) In Fig. S1, the yield of Xu8 is comparable to that of Xu1, yet the enantioselectivity is significantly higher. What prompted the modification of the catalyst structure based on Xu1 rather than Xu8?

*Reply: That's an excellent question and we're sure that some readers also have the same question. Our relevant precedents (Acc. Chem. Res. 2024, 57, 489), along with this study, have once again demonstrated that Xu1 (3,5-di-*t*-butyl-4-methoxyphenyl, DTBM) is more universally applicable in Xu-Phos. This observation is the primary rationale behind our decision to modify the ligand structure based on Xu1 rather than Xu8. The results were discussed in the revised Supplementary Information. (SI, Page S6)*

(5) Is it necessary for flexible ethylene halide 1 to bear substituent groups in Fig. 3? Have the authors experimented with substrates lacking substituents, and if so, what were the results? Can the authors produce seven-membered ring products? The authors posit that stereoselectivity arises from the Heck reaction step. Have they tried using acyclodienes? Can enantioselectivity and diastereoselectivity be controlled in such cases?

Reply: According to your nice suggestions, we conducted the reactions of ambiphilic halogenated allylic alcohols S-1a, which are unsubstituted (● = H), with cyclic 1,3-dienes 2a under the standard conditions. However, the corresponding product was not obtained with the S-1a (● = H). This may be due to the cis-trisubstituted alkenes exhibit greater stability compared to cis-disubstituted alkenes in terms of cis/trans isomerization. And the substrates trans-1 has high stability and low reactivity, which prevents the formation of the five-membered palladacycles intermediate (Fig. 5b in manuscript).

Then according to the reviewer's suggestion, we also conducted the reaction of model flexible halogenated allylic alcohol **1a** with cyclopentadienes or cycloheptadienes under the standard conditions to provide a better sense of the limitations of the substrate scope. For the cyclopentadiene **S-2c**, the corresponding five-membered ring product **S-3e** was obtained in 9% yield and with 6% ee, due to the cyclopentadiene easily undergo dimerization ([4+2]cycloaddition). However, the corresponding seven-membered ring product **S-3d** was not obtained with the cycloheptadienes **S-2b**.

The reaction with acyclic 1,3-diene **S-2d** work, but very low ee was obtained, indicating that the Tsuji-Trost step is not the enantio-determining step. These results suggest that we have a long way to address the remained issues and the further design and development of new ligand is still highly desirable. These results and date have been added to the revised Supplementary Information. (Page S6)

- (6) In the HPLC analysis, the racemate of compound **5** does not appear as a single compound. Additionally, in the NMR spectra, the solvent peaks for compounds **3ba** and **4** are excessively high.

*Reply: Thank you for careful review. We have carefully corrected the errors. The new HPLC analysis of compound **5** and the NMR spectra **3ba** and **4** have been added to the revised Supplementary Information. (Page S26, S44; S45, S77, S103)*

- (7) All 2D spectra lack peaks and resolution.

Reply: Thank you for pointing this out. All analytical 2D spectra have been added to the revised Supplementary Information. (Page S103—S121)

Reply to the comments and suggestions of the reviewer #2:

The authors describe an asymmetric heteroannulation reaction of vinylic halides with 1,3-cyclohexadienes. This reaction makes use of chiral phosphine/sulfinimide ligands, which are effective at inducing asymmetry in the reaction with cyclohexadienes. The authors show that acyclic dienes are ineffective in the reaction. Among the vinylic halides, a range of substitution works well in the 2-position, including both electron-rich and electron-poor aryl substitution and alkyl groups. A similar range of substitution is well-tolerated among the cyclohexadienes, although in neither case is there variation in the position of substituents, and the demonstration of functional group tolerance is largely lacking (the authors show that thiophene, benzothiophene work, but no nitrogen functionality is present throughout the scope). The reaction scales with comparable reactivity (79% vs. 83% yield), and the authors show some typical reactions for elaboration of the alkenes in the

products. An asymmetric induction model is provided, although without experimental support.

Overall, this is a nice demonstration of asymmetric catalysis for the synthesis of interesting products. For the scope, the e.e. values are generally quite good, with a decent scope, and they map directly onto bioactive scaffolds of interest. While there have been several reports of enantioselective heteroannulation, this is to my knowledge the first with non-aromatic bifunctional reagents, which is a nice synthetic advance. However, despite the number of substrates, overall the substrate scope is narrow. There is limited demonstration of functional group tolerance, and it is unclear whether the method can be extended to other cyclic dienes. The SI is generally good and quite detailed, but there are some issues with the reported HPLC traces that need to be addressed (see below for details). Also, the authors should indicate both in the substrate scope tables and SI that the yields reported represent single runs and have not been reproduced.

Questions and Suggestions:

- 1) Page 1, line 40-41: "seminal researchs" is misspelled, and would be more appropriate to say "seminal reports;" also the sentence appears to be a fragment and not a complete sentence as written. If the authors intend to use periods between the numbered items in this paragraph, they should capitalize the first word and make sure that they stand alone as sentences. I personally think it would be sufficient to use a semicolon between the numbered items.

Reply: We thank you for the high comments and good suggestions, which encourages us to conduct more comprehensive and systematic domino Heck/Tsuji-Trost reaction. According to your nice suggestions, we have rectified the inaccuracies in the manuscript and Supplementary Information. And "seminal researchs" was revised as "seminal reports"; we have used a semicolon between the numbered items. (Page 1, line 40-41). We have clearly indicated both in the substrate scope tables and SI that the yields reported represent single runs and have not been reproduced. (Fig. 3 and 4; and Note of SI)

- 2) Fig. 1A-B: The wording " η^3 π -allyl" is redundant; the authors should either say " η^3 allyl" or " π -allyl"

Reply: Thank you for pointing this out. We have rectified the inaccuracies in the manuscript by employing the " η^3 allyl" notation in place of the " η^3 π -allyl".

- 3) Fig. 4: From Fig. 2b, it seems that the reaction is limited to cyclic dienes for the purposes of asymmetric induction. However, the authors only show 1,3-cyclohexadienes in their scope, and there is no evidence of other cyclic dienes in the SI either. Will this reaction work with cyclopentadienes or cycloheptadienes? The authors should include these results to provide a better sense of the limitations (or potential diversity) of the substrate scope.

*Reply: T we also conducted the reaction of model flexible halogenated allylic alcohol **1a** with cyclopentadienes or cycloheptadienes under the standard conditions to provide a better sense of the limitations of the substrate scope. For the cyclopentadiene **S-2c**, the corresponding five-membered ring product **S-3e** was obtained in 9% yield and with 6% ee, due to the cyclopentadiene easily undergo dimerization([4+2]cycloaddition). However, the corresponding seven-membered ring product **S-3d** was not obtained with the cycloheptadienes **S-2b**.*

*The reaction with acyclic 1,3-diene **S-2d** work, but very low ee was obtained, indicating that the Tsuji-Trost step is not the enantio-determining step. These results suggest that we have a long way*

to address the remained issues and the further design and development of new ligand is still highly desirable. These results and data have been added to the revised Supplementary Information. (Page S6)

- 4) Fig. 5B: The authors propose a model for asymmetric induction, but without any direct experimental or computational support. They rely on prior work with different systems (most of the provided models in the ACR paper use olefins as the bound substrates, and many of those are thin on experimental support as well). It would be helpful for the authors to either provide more support for their proposed model (either through additional experimentation/computational study) or to use “plausible asymmetric induction model” instead of “proposed.” In reality, this contributes little to what is otherwise an entirely synthetic paper, and could be left out.

Reply: According to your nice suggestions, we use “plausible asymmetric induction model” instead of “proposed” in the manuscript (Fig. 5B). And the sentence “a catalytic chirality-induction model……” was revised as “a possible catalytic chirality-induction model ……”. Currently, we are preparing the DFT calculations of the asymmetrical version on the enantio-determining step, this part will be published separately because of deadline of revision due.

- 5) References: While the authors’ focus is on asymmetric catalysis, it would be highly relevant for them to include references to work from the Richard Larock group and Keary Engle group on olefin heteroannulation reactions with non-aromatic bifunctional coupling partners, which provide reactivity precedent for the authors’ present asymmetric catalysis work with these types of coupling partners.

Reply: According to your nice suggestions, we have added these references to cite. (Engle, Mapping Amphiphile Reactivity Trends in the Anti-(Hetero)annulation of Non-Conjugated Alkenes via Pd^{II}/Pd^{IV} Catalysis. Angew. Chem. Int. Ed. 2022, 61, e202114346; Larock, Palladium-Catalyzed Annulation of Allenes Using Functionally Substituted Vinylic Halides. J. Org. Chem. 1998, 63, 2154–2160 (ref. 44, 45).

- 6) SI: Some of the racemic HPLC traces are not clean, with multiple impurities present near the peaks of interest; this could represent impurities that lie under the peaks of interest in the HPLC traces of enantioenriched product, leading to inaccuracies in the reported e.e. values. The authors should address these discrepancies. The relevant compounds are: 3ah, 3ak, 3bk (in this case, the peak integration of the racemic sample is 48:52, outside the acceptable error for a pure, racemic sample), 3bp, 3bs (peak integration of racemic sample is 43:57, with clear impurity

peaks under the desired peaks)

Reply: According to your nice suggestions, we have repurified the HPLC traces and data, and now the 3ah, 3ak, 3an, 3bk, 3bp, 3bs is correct (SI, Page S15, S17, S19, S33, S37, S39)

- 7) SI: For compound 3an, there is significant overlap between the enantiomer peaks in the HPLC traces, making it very difficult to be confident in the integrations (and thus, reported e.e. values) of the enantioenriched traces. The authors should identify a different separation method in order to properly resolve the peaks of interest, and update their reported e.e. values accordingly.

Reply: Thank you for careful review. We have employed Chiralpak IF column (n-Hexane:2-Propanol = 95:5, 0.5 ml/min) to enhance chiral separation and updated the e.e. values. (SI, Page S19)

Reply to the comments and suggestions of the reviewer #3:

In this manuscript, Junliang Zhang and coworkers describe an enantioselective Heck/Tsuji-Trost domino reaction. The novelty of this protocol is firstly the incorporation of acyclic, flexible vinyl halides as substrates. Secondly, it is the Heck carbometalation that forms the stereo-determining step (the source of enantioselectivity) rather than the Tsuji-Trost allylation (that is widely explored). These are two quite remarkable points. High degrees of enantioselectivity are obtained and the versatility is documented by a variety of substrates.

Despite the progress the authors made in “acyclic stereoselection”, a final challenge remains to be tackled (in the future): the step from the cyclic 1,3-diene to an acyclic 1,5-disubstituted 1,4-diene! Nevertheless, the manuscript at hand discloses an innovative, valuable procedure. Therefore, I strongly recommend acceptance of the manuscript.

Some alterations are necessary:

- 1) The term “tandem” is not adequate in this context and should be replaced by “domino”, as explained by Lutz Tietze. In a tandem, a simultaneous motion occurs. This is not the case in the reactions at hand, where one step is prerequisite to the next. I also recommend citation of the comprehensive overviews: L.-F. Tietze, G. Brasche, K. M. Gericke, Domino Reactions in Organic Synthesis, Wiley-VCH: Weinheim, Germany 2006; b) L.-F. Tietze (Ed.), Domino Reactions: Concepts for Efficient Organic Synthesis, Wiley-VCH, Weinheim, Germany 2014.

Reply: We thank you for the high comments and good suggestions, which encourages us to conduct more comprehensive and systematic domino Heck/Tsuji-Trost reaction. According to your nice suggestions and careful review, we have rectified the inaccuracies in the manuscript: “domino” instead of “tandem”; I sincerely apologize for the oversight, we have added these representative references to cite. (ref. 4 and 5)

- 2) To me, parts of Fig. 1 are not clear. Fig. 1a, the second line simply illustrates the Tsuji-Trost reaction. Why is ref. 17 given here – instead of citations of comprehensive reviews by Barry Trost?

Reply: Thank you for pointing this out. We have carefully checked the typos and corrected the above errors in Manuscript (Fig.1). And “ref. 17” was revised as “ref. 35”

- 3) Also Fig. 1b: Where are fragments and residue R (second line, in brackets) in the final product??

Reply: Thank you for pointing this out. We have carefully checked the typos and corrected the above errors in Manuscript (Fig. 1b). And we have redrawn the Fig 1b, “R” was revised as “dotted line”.

4) Abstract: line 18 should read “as” instead of “of” (stereodetermining step).

Reply: Thank you for pointing this out. We have carefully checked the typos and corrected the above errors in Manuscript (Abstract: line 18). And “of” was revised as “as”.

5) page 5: Fig.4 and line 114: menthol and peryllic alcohol (no capitals)

Reply: Thank you for pointing this out. We have carefully checked the typos and corrected the above errors in Manuscript. (page 5: Fig.4 and line 114). And “Menthol and Peryllic alcohol” was revised as “menthol and peryllic alcohol”.

In summary: I recommend acceptance of the manuscript after alterations, additions and corrections as outlined above.

Reply: Thank you for your kind suggestions, which help us to improve the quality of this work.